# Detection and Monitoring of Highly Pathogenic Influenza A Virus 2.3.4.4b Outbreak in Dairy Cattle in the United States

**DOI:** 10.3390/v16091376

**Published:** 2024-08-29

**Authors:** Luis G. Giménez-Lirola, Brooklyn Cauwels, Juan Carlos Mora-Díaz, Ronaldo Magtoto, Jesús Hernández, Maritza Cordero-Ortiz, Rahul K. Nelli, Patrick J. Gorden, Drew R. Magstadt, David H. Baum

**Affiliations:** 1College of Veterinary Medicine, Iowa State University, Ames, IA 50011, USA; bmhend14@iastate.edu (B.C.); juanmora@iastate.edu (J.C.M.-D.); rmagtoto@iastate.edu (R.M.); rknelli@iastate.edu (R.K.N.); pgorden@iastate.edu (P.J.G.); magstadt@iastate.edu (D.R.M.); dhbaum@iastate.edu (D.H.B.); 2Laboratorio de Inmunología, Centro de Investigación en Alimentación y Desarrollo, A.C. Hermosillo, Sonora 83304, Mexico; jhdez@ciad.mx (J.H.); maritzacoordero@gmail.com (M.C.-O.)

**Keywords:** highly pathogenic influenza A virus, H5N1 2.3.4.4b, outbreak, dairy cattle, seroconversion, acute phase, convalescent phase, blocking ELISA, diagnostic performance

## Abstract

The emergence and spread of highly pathogenic avian influenza virus A subtype H5N1 (HP H5N1-IAV), particularly clade H5N1 2.3.4.4b, pose a severe global health threat, affecting various species, including mammals. Historically, cattle have been considered less susceptible to IAV, but recent outbreaks of H5N1-IAV 2.3.4.4b in dairy farms suggest a shift in host tropism, underscoring the urgency of expanded surveillance and the need for adaptable diagnostic tools in outbreak management. This study investigated the presence of anti-nucleoprotein (NP) antibodies in serum and milk and viral RNA in milk on dairy farms affected by outbreaks in Texas, Kansas, and Michigan using a multi-species IAV ELISA and RT-qPCR. The analysis of ELISA results from a Michigan dairy farm outbreak demonstrated a positive correlation between paired serum and milk sample results, confirming the reliability of both specimen types. Our findings also revealed high diagnostic performance during the convalescent phase (up to 96%), further improving sensitivity through serial sampling. Additionally, the evaluation of diagnostic specificity using serum and milk samples from IAV-free farms showed an excellent performance (99.6%). This study underscores the efficacy of the IAV NP-blocking ELISA for detecting and monitoring H5N1-IAV 2.3.4.4b circulation in dairy farms, whose recent emergence raises significant animal welfare and zoonotic concerns, necessitating expanded surveillance efforts.

## 1. Introduction

The highly pathogenic avian influenza virus A subtype H5N1 (HP H5N1-IAV), first reported in 1996 in Guangdong, China [1], has evolved into clade H5N1 2.3.4.4b, recently infecting various species worldwide, including mammals [2,3,4]. The characteristics of this virus and its ability to transmit to mammals underline its pandemic potential [5], especially with recent confirmations of H5N1-IAV in U.S. dairy cattle (Texas, Kansas, Michigan, Idaho, New Mexico, North Carolina, Ohio, and South Dakota) [6].

The emergence of H5N1-IAV 2.3.4.4b in cattle has raised significant concerns regarding zoonotic transmission and its potential for triggering a pandemic. Historically, cattle have been considered less susceptible to IAV, with only sporadic descriptions of dead-end transmissions [7], but recent outbreaks in dairy farms suggest a shift in host tropism. This shift underscores the pressing need for enhanced surveillance and containment efforts [5].

Thus, there is an urgent need for high-throughput, cost-effective testing tools for outbreak investigations, such as antibody assessment. Through accurate detection of antibodies within affected animal populations, we not only elucidate disease transmission dynamics but also enhance our ability to implement tailored interventions. This approach helps to safeguard animal welfare and protect vulnerable cohorts, highlighting the vital role of antibody testing in mitigating the spread and impact of this emerging disease through a fast and simple test.

When a new disease emerges in a population, the first step should be assessing the efficacy of existing diagnostic tools. Therefore, a nested case–control serodiagnostic accuracy assessment was conducted within a dairy cattle cohort exposed to IAV H5N1 outbreaks in Texas, Kansas, and Michigan. As part of this study protocol, weekly serum and milk samples were collected from the cattle for IAV nucleoprotein (NP)-blocking ELISA testing.

## 2. Materials and Methods

### 2.1. Samples

Serum (*n* = 161) and milk (*n* = 103) samples were collected between 16 March and 17 April 2024, from dairy cattle farms in Texas, Kansas, and Michigan (Table 1). These farms had suspected cases of H5N1 2.3.4.4b based on clinical findings such as low appetite, reduced milk production, and abnormal milk appearance, and virological laboratory confirmation using quantitative reverse transcription polymerase chain reaction (RT-qPCR), immunohistochemistry, and sequencing methods [8,9]. Additionally, serum (*n* = 371) and milk (*n* = 100) samples were collected from IAV-free farms in Iowa, Montana, South Dakota, and Texas during the Spring of 2024 (Table 1).

### 2.2. Real-Time PCR

Milk samples were diluted in PBS or molecular transport medium (MTM) at a ratio of 1:3. A volume of 200 μL of sample in a deep-well plate was used for nucleic acid extractions on KingFisher Flex (Thermo Fisher Scientific, Waltham, MA, USA) using MagMax Pathogen RNA/DNA Kit as per the National Veterinary Services Laboratories (NVSLs) guidelines. All reverse-transcription quantitative PCR (RT-qPCR) assays were performed using a National Animal Health Laboratory Network (NAHLN)-approved assay [10] with a modification to use the VetMAX-Gold SIV Detection kit (Thermo Fisher Scientific, Waltham, MA, USA) to screen for the presence of influenza A virus RNA. All tested samples included the VetMAX XENO Internal Positive Control to monitor the possible presence of PCR inhibitors. Each RT-qPCR 96-well plate had 2 positive amplification controls, 2 negative amplification controls, 1 positive extraction control, and 1 negative extraction control. After the RT-qPCR screening, positive samples for IAV RNA were further tested for the H5 subtype and H5 clade 2.3.4.4b using the same RNA extraction and NAHLN-approved RT-qPCR protocols according to standard operating procedures. The assays were performed on the ABI 7500 Fast thermocycler, and data were analyzed using Design and Analysis Software 2.7.0 (Thermo Fisher Scientific). Samples with cycle threshold (Ct) values < 40.0 were considered to be positive for IAV and its H5 subtype.

### 2.3. Antibody Detection in Serum and Milk Samples

Anti-influenza antibodies to monitor H5N1-IAV 2.3.4.4b outbreaks in dairy cattle were evaluated with a commercial IAV enzyme-linked immunosorbent assay (ELISA) (Cat# 99-0000900; Idexx Laboratories Inc., Westbrook, ME, USA) designed to detect anti-NP antibodies in multiple species.

Antibody testing was performed on both bovine serum and defatted milk according to the manufacturer’s instructions, with minor modifications. Specifically, serum samples were tested at a 1:10 dilution while defatted milk samples were tested undiluted. Milk samples were first centrifuged at 13,000× *g* for 15 min at 4 °C to remove the upper fat layer. Then, 5 µL of Rennet (Rennet from Mucor miehei; Millipore-Sigma, Burlington, MA, USA) stock solution (0.5 g/mL in sterile water) per mL of milk was added, vortexed for ~30 s, and incubated at 37 °C for 30 min. After centrifugation at 2000× *g* for 15 min, the clear fluid (serum portion) between the curve and fat layers was collected for antibody testing.

Undiluted ready-to-use kit positive controls (PCs) and negative controls (NCs) (100 µL per well) were run in duplicate wells. After 60 min incubation at room temperature (RT; 22–24 °C), each well was washed 5 times with 350 µL of wash solution, avoiding plate drying between plate washings and prior to the addition of the next reagent. The plate was tapped onto a paper towel after the final wash to remove residual wash fluid. Then, 100 µL of conjugate was added into each well and incubated for 30 min at RT. After another washing step, 100 µL of 3,3′,5,5′-Tetramethylbenzidine (TMB) substrate solution was added to each well and incubated for 15 min at RT. The reaction was stopped by adding 100 µL of stop solution into each well. In the blocking ELISA method, anti-NP antibodies present in the sample block the binding of the anti-NP conjugate to the NP antigen on the plate. Color development is inversely proportional to the quantity of IAV antibodies in the test sample, calculated as a sample-to-negative (S/N) value as follows:$$S/N=\frac{{Sample\;Absorbance\;(A)}_{650}}{{NC\;mean\;A}_{650}}$$

According to the manufacturer’s instructions, samples with an S/N value < 0.6 were considered positive.

### 2.4. Statistical Analysis

The distribution of IAV NP ELISA S/N values in serum samples collected over the course of the infection (pre-clinical, acute, and convalescent phases) with H5N1-IAV 2.3.4.4b were compared using a one-way ANOVA model, followed by post hoc tests to assess differences between pairs of stages. The *p*-values of the post hoc tests were adjusted using Tukey’s method to control a familywise error rate of 0.05. A Welch two-sample *t*-test was performed to determine the difference in positive or negative detection of antibodies against IAV NP within each phase of the disease.

Variations in the detection of IAV NP antibodies in paired serum samples obtained from dairy cattle during the acute and chronic phases of H5N1-IAV 2.3.4.4b infection were assessed using a paired *t*-test. The statistical significance between IAV NP ELISA S/N values obtained from testing serum and milk specimens from both IAV-free and H5N1-IAV 2.3.4.4b-affected dairy cattle during the convalescent phase of infection was determined via two-way ANOVA with Šídák’s multiple comparisons test. Additionally, a Pearson correlation coefficient was calculated for qualitative results obtained from paired serum and milk samples. In all analyses, a *p*-value < 0.05 was considered statistically significant. Statistical analyses were performed using RStudio 2024.04.0+735 (SAS Institute, Cary, NC, USA). Graphical representation of the data was performed using GraphPad Prism^®^ 10.2.3 (GraphPad Software Inc., San Diego, CA, USA).

## 3. Results

The present study encompasses a serological investigation of the HP H5N1-IAV 2.3.4.4b outbreak in a Michigan dairy farm. A comprehensive panel of paired serum and milk samples was systematically collected from 27 animals during the convalescent phase of infection. Milk samples were tested by RT-qPCR, with Ct values ranging from 27.3 to 38.5, at the Veterinary Diagnostic Laboratory of Iowa State University, and all samples were positive. Serum samples were not tested by RT-PCR due to the normal absence of viremia. Multi-species blocking IAV ELISA detecting anti-NP antibodies showed a detection rate of 96% (26 of 27) in serum and 89% (24 of 27) in milk. These results showed a positive correlation (R^2^ = 0.93), with only one animal being seronegative for IAV in both serum and milk and two animals testing positive in serum but negative in milk (Figure 1). The analysis also included a panel of samples from IAV-free dairy and beef cattle farms across multiple states, including Iowa, Montana, South Dakota, and Texas: 371 negative serum samples and 100 paired serum and milk samples. With the exception of one serum sample, all serum and milk negative samples were found to be negative by the ELISA, showing a diagnostic specificity of 99.6%. The only “false-positive” serum (Iowa beef cattle farm) had an S/N value of 0.58, just below the kit manufacturer cutoff value of 0.6 (Figure 1). These results show that multi-species anti-influenza IAV-blocking ELISA is a suitable method for detecting antibodies anti-IAV in serum and milk from cows infected with the HP H5N1-IAV.

Then, the presence of anti-IAV antibodies in milk was evaluated. To better understand the accuracy and power of this assay, serial milk samples (four samplings taken one week apart) collected from 19 dairy cattle from the original H5N1-IAV outbreak reported in Texas were evaluated. Table 2 shows the dynamics of the anti-IAV antibodies and the virus in milk. At week 0, when the clinical signs in the cows were evident (i.e., low appetite, reduced milk production, and abnormal milk appearance) [11], 16 of 19 milk samples were PCR-positive, and only 5 of 19 showed anti-IAV antibodies. Four of the nineteen milk samples with anti-IAV antibodies were also PCR-positive. At week 1, 14 of 19 milk samples were PCR-positive, and 15 of 19 milk samples had anti-IAV antibodies. Eleven out of nineteen milk samples with anti-IAV antibodies were also PCR-positive. In week 2, 11 of 19 milk samples remained PCR-positive and 17 of 19 showed IAV antibodies. Ten out of nineteen milk samples with anti-IAV antibodies were also PCR-positive. At week 4, all milk samples were PCR-negative, and 18 of 19 had anti-IAV antibodies. During this analysis (week 4), one sample (ID 12) was negative for anti-IAV antibodies and was only PCR-positive in week 2. Based on this, Table 3 confirms the accuracy of this assay 1 week after the onset of clinical signs. These results showed that the detection of anti-IAV antibodies in milk from one week since the beginning of the outbreak can be used as an indicator of HP H5N1-IAV infection.

Then, serum samples were collected from 66 animals from a dairy farm in Kansas at various stages throughout the infection period (Table 1). The detection rate of anti-IAV antibodies using multi-species blocking ELISA across different phases of infection was evaluated (Figure 2). The results showed that the detection rate increased over time, with no detections before the onset of clinical signs (pre-clinical phase), followed by increased detection during the acute phase (within the first 7 days after onset of clinical signs; 9 of 24; 39%), and reaching its highest level during the convalescent phase (14 days after the onset of clinical signs; 23 of 25; 92%). The post hoc pairwise comparisons showed significant variations in S/N values among samples collected during the pre-clinical, acute, and convalescent phases of infection (*p* < 0.0001) (Figure 2). Furthermore, *t*-tests unveiled a statistically significant difference (*p* < 0.0001) between seropositive or seronegative animals sampled during the acute phase of infection, while such differences were not significant within the pre-clinical and convalescent groups (*p* > 0.05) (Figure 2).

The diagnostic sensitivity of this antibody test was below 40% during the first week following the onset of the clinical signs, limiting its primary diagnostic role. However, the diagnostic value of serial sampling from the same animals during the acute and convalescent phases of infection was demonstrated using serum samples taken one week apart from H5N1-IAV 2.3.4.4b outbreaks in dairy farms in Kansas (22 animals; 44 serum samples) and Michigan (12 animals; 24 serum samples). Overall, the diagnostic detection rate increased from 29% (10/34) during the acute phase to 94% (32/34) during the convalescent phase (Figure 3).

## 4. Discussion

This study demonstrates the effectiveness of an IAV NP-blocking ELISA for detecting and monitoring HP H5N1-IAV 2.3.4.4b circulation in dairy farms, in serum, and in milk specimens. Since the standardization of a new serologic test for IAV in cattle would be time-consuming, the utility of this ELISA test turns out to be particularly evident during outbreaks, especially in scenarios with low prevalence, thus facilitating containment efforts. Also, even though the use of RT-qPCR for the diagnosis of the disease is a gold-standard technique, the time span for detection could be short, which would represent a problem in asymptomatic infections. This study did not aim to evaluate the diagnostic sensitivity and specificity of the multi-species IAV NP-blocking ELISA. However, an analysis of almost 400 AIV-free serum samples revealed only one false positive, demonstrating the high diagnostic specificity of the test in cows. The use of a multi-species NP ELISA to detect IAV in cattle is justified, as cattle were historically resistant to IAV, with few cases until the emergence of influenza D and highly pathogenic IAV strains. The highly conserved NP protein allows cross-species detection, making the assay practical, cost-effective, and reliable for this newly recognized host.

The detection rate of PCR-positive samples was high: 96% and 89% for serum and milk, respectively, confirming the power of this assay in detecting antibodies in cows infected with the HP H5N1-IAV 2.3.4.4b virus. Furthermore, we are developing novel methods to enhance our diagnostic capabilities, including H5-specific immunoassays. Collaborative efforts across veterinary, agricultural, and public health sectors are imperative for mitigating transmission risks and safeguarding both animal and human health.

The presence of the virus in the milk of the affected cows is an important public health concern. This study evaluated the dynamics of antibodies anti-AIV and the virus over four weeks using paired milk samples. Notably, the seroconversion was evident in some cows at week 0 (the time of symptoms onset) and increased each week until week 4, where 18 of 19 cows were positive. When comparing the dynamic of the virus and the dynamic of the antibodies, in week 4, none of the milk samples were PCR-positive. These results indicate that the presence of the virus in milk is evident in the first three weeks after the symptoms onset, but after four weeks, it is improbable to detect the virus in milk. Also, it suggests that the detection of anti-AIV antibodies in milk can be an indicator of infection, especially in cases where PCR testing is not performed in the first few weeks after the outbreak. An important question arising in this study is as follows: what is the duration of these antibodies in milk? Further studies are needed to evaluate the duration of antibodies in milk.

Animals infected with the HP H5N1-IAV had very short viremia, usually less than one week. No studies have demonstrated a similar scenario in cows infected with the HP H5N1-IAV 2.3.4.4b, but the current diagnostic observations seem to confirm this scenario [12]. In contrast, antibody detection in sera is common in animals infected with the influenza virus [13]. This study compared the antibody response in cows at three different points: pre-clinical, acute, and convalescent phases. As expected, the seroconversion was more evident at the convalescent stage than in the acute phase.

## 5. Conclusions

The results of this study demonstrate that the antibody response in cows infected with the HP H5N1-IAV 2.3.4.4b can be effectively detected using a multi-species NP-based blocking ELISA. Antibodies are detectable in the serum of cows with acute disease, but the detection rate is higher in convalescent animals. This ELISA can be a valuable tool for assessing the spread of the disease and conducting seroepidemiological studies. Additionally, the assay is capable of detecting antibodies in milk, with a good correlation to the presence of virus, particularly 1 or 2 weeks after the onset of the clinical signs. A limitation of this study is the inability to evaluate the presence of antibodies in serum and milk over extended periods, as the exact time of infection could not be determined.

## Figures and Tables

**Figure 1 viruses-16-01376-f001:**
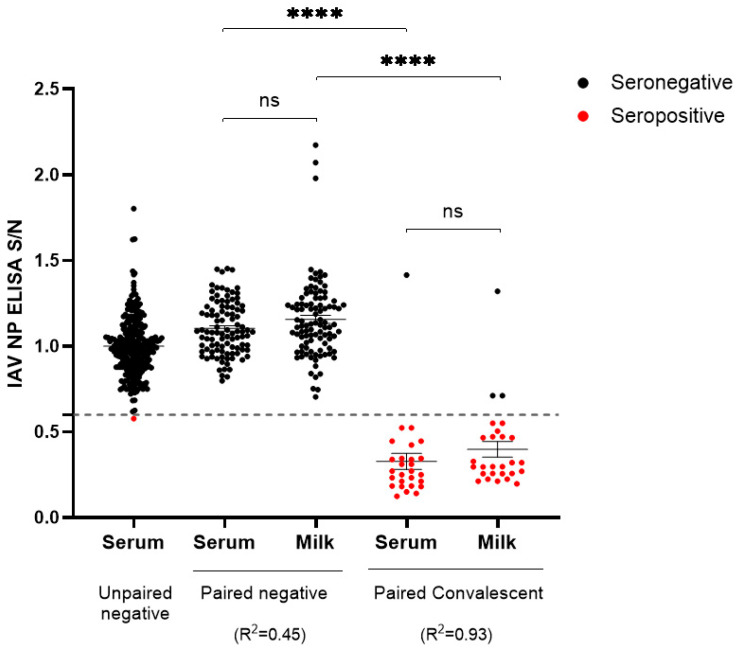
Evaluation of anti-NP antibodies in positive and negative serum and milk samples. Distribution of positive (denoted by red dots) and negative (denoted by black dots) influenza A virus (IAV) nucleoprotein (NP)-blocking ELISA sample-to-negative (N/S) results using an S/N cutoff of 0.6 (represented by the dashed line). Data correspond to testing of known IAV-negative serum (*n* = 371) and paired serum and milk (*n* = 100) samples, alongside convalescent serum (*n* = 27) and milk (*n* = 27) samples from HP H5N1-IAV 2.3.4.4b-infected dairy cattle. The statistical significance was determined using 2-way ANOVA with Šídák’s multiple comparisons test. The Pearson correlation coefficient was calculated for paired samples. **** denotes a *p*-value of <0.0001, while “ns” denotes non-statistically significant difference.

**Figure 2 viruses-16-01376-f002:**
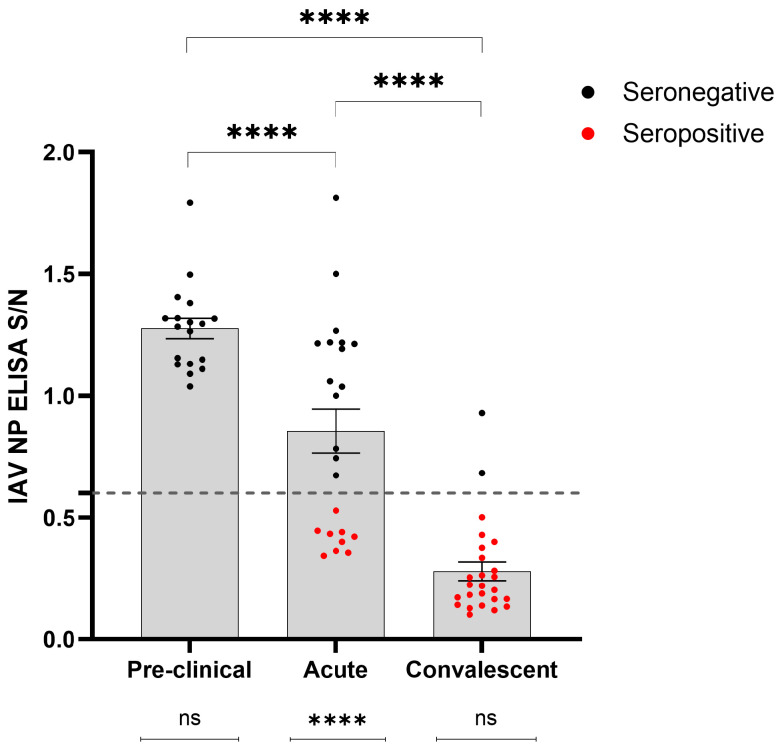
Detection of influenza A virus (IAV) nucleoprotein (NP) antibodies in serum samples collected from 66 dairy cattle in Kansas during the pre-clinical, acute, and convalescent phases of infection with H5N1-IAV 2.3.4.4b. Samples with sample-to-negative (S/N) values below the 0.6 cutoff (represented by the dashed line) were classified as positive (denoted by red dots), while those above were classified as negative (denoted by black dots). A one-way ANOVA model followed by post hoc pairwise tests was used to assess differences between the pair of stages. A Welch two-sample *t*-test was performed to assess the differences in positive or negative detection of antibodies against IAV NP within each phase of the disease. **** denotes a *p*-value < 0.0001, and “ns” denotes no significant difference.

**Figure 3 viruses-16-01376-f003:**
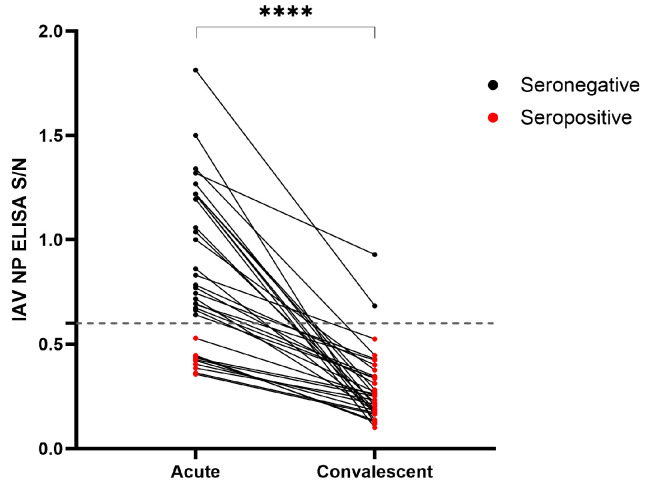
Detection of influenza A virus (IAV) nucleoprotein (NP) antibodies in paired serum samples collected from 34 dairy cattle during acute (*n* = 68) and chronic phases (*n* = 68) of infection with H5N1-IAV 2.3.4.4b. Samples with sample-to-negative (S/N) values below the 0.6 cutoff (represented by the dashed line) were classified as positive (denoted by red dots), while those above the cutoff were classified as negative (denoted by black dots). The statistical significance was determined using the paired *t*-test. **** denotes a *p*-value < 0.0001.

**Table 1 viruses-16-01376-t001:** Description of serum and milk samples collected from both highly pathogenic influenza A virus (H5N1-IAV) 2.3.4.4b-affected farms and IAV-free farms included in this study.

Premise	Location	Specimen	No. Animals	No. Samples	Sampling Procedure	Status
Dairy cattle	Kansas	Serum	66	66	Same site, serial (not paired) sampling	H5N1-IAV-affected; pre-clinical, acute and convalescent phas
Dairy cattle	Kansas	Serum	22	44	Paired sampling	H5N1-IAV-affected; acute and convalescent phase
Dairy cattle	Michigan	Serum	12	24	Paired sampling	H5N1-IAV-affected; acute and convalescent phase
Dairy cattle	Michigan	Serum/milk	27	54	Paired specimens	H5N1-IAV-affected; convalescent phase
Dairy cattle	Texas	Milk	19	76	Serial sampling	H5N1-IAV-affected; acute through convalescent phase
Dairy cattle	Iowa	Serum/milk	100	200	Paired specimens	IAV-free
Dairy cattle	Iowa	Serum	43	43	Cross-sectional	IAV-free
Beef cattle	Iowa	Serum	131	131	Cross-sectional	IAV-free
Beef cattle	Montana	Serum	72	72	Cross-sectional	IAV-free
Beef cattle	South Dakota	Serum	15	15	Cross-sectional	IAV-free
Beef cattle	Texas	Serum	10	10	Cross-sectional	IAV-free

**Table 2 viruses-16-01376-t002:** Detection of viral RNA (RT-qPCR) and antibodies against the nucleoprotein (NP) of influenza A virus (IAV blocking ELISA; Idexx Laboratories) in milk samples serially collected post clinical onset from Texas dairy cattle premises (*n* = 19) affected by highly pathogenic influenza A virus (H5N1-IAV) 2.3.4.4b.

	IAV NP-Blocking ELISA and RT-qPCR Results										
	Week 0 *			Week 1			Week 2			Week 4	
ID	S/N	PCR		S/N	PCR		S/N	PCR		S/N	PCR
1	2.481	19.8		**0.243**	29.8		**0.136**	>40		**0.182**	>40
2	0.649	22.9		0.612	28		**0.268**	>40		**0.206**	>40
3	1.391	22.5		**0.342**	31.6		**0.246**	36.6		**0.213**	>40
4	1.926	21.9		**0.236**	28.5		**0.162**	38.5		**0.303**	>40
5	1.260	22.2		0.961	30.1		**0.186**	35.8		**0.387**	>40
6	**0.255**	30.7		**0.277**	>40		**0.177**	32.4		**0.198**	>40
7	2.032	21.9		**0.482**	29.8		**0.204**	35.3		**0.165**	>40
8	**0.353**	26.4		**0.298**	27.3		**0.197**	36.4		**0.171**	>40
9	0.744	26.7		**0.341**	31.8		**0.176**	>40		**0.159**	>40
10	1.124	27.5		0.951	35		0.635	>40		**0.363**	>40
11	0.938	23		**0.317**	27.9		**0.182**	>40		**0.245**	>40
12	1.265	>40		1.055	>40		0.756	31.6		0.645	>40
13	0.619	>40		**0.196**	>40		**0.213**	>40		**0.346**	>40
14	1.733	20.4		**0.265**	33.3		**0.246**	33		**0.164**	>40
15	**0.346**	<40		**0.340**	>40		**0.231**	>40		**0.262**	>40
16	1.796	22.2		**0.344**	28.3		**0.185**	35.8		**0.133**	>40
17	**0.315**	25.4		**0.267**	35.6		**0.191**	>40		**0.197**	>40
18	**0.461**	29.4		**0.367**	>40		**0.257**	35.6		**0.184**	>40
19	1.241	21.3		**0.351**	28.1		**0.219**	33.8		**0.225**	>40

* Samples were collected during the acute phase of infection. Bold letters indicate positiveness (values below the cutoff, 0.6).

**Table 3 viruses-16-01376-t003:** Contingency table of IAV NP-blocking ELISA and RT-qPCR results (controls) in milk samples (*n* = 76) from each week, collected post clinical onset from Texas dairy cattle premises affected by highly pathogenic influenza A virus (H5N1-IAV) 2.3.4.4b.

Week 0				Week 2			
	RT-PCR (+)	RT-PCR (−)	Total		RT-PCR (+)	RT-PCR (−)	Total
ELISA (+)	4	1	5	ELISA (+)	10	7	17
ELISA (−)	12	2	14	ELISA (−)	1	1	2
Total	16	3	19	Total	11	8	19
**Week 1**				**Week 4**			
	RT-PCR (+)	RT-PCR (−)	Total		RT-PCR (+)	RT-PCR (−)	Total
ELISA (+)	11	4	15	ELISA (+)	0	18	18
ELISA (−)	3	1	4	ELISA (−)	0	1	1
Total	14	5	19	Total	0	19	19

## Data Availability

Data are contained within the article.

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
