# Peer review of "Detection and Monitoring of Highly Pathogenic Influenza A Virus 2.3.4.4b Outbreak in Dairy Cattle in the United States"

_viruses, 2024, doi:10.3390/v16091376_

Round 1

Reviewer 1 Report

Comments and Suggestions for Authors

 Line 73-74 update to “National Veterinary Services Laboratories (NVSL)”

·         Line 75-76 update to “National Animal Health Laboratory Network (NAHLN)”

·         Edit line 244 to “HP H5N1-IAV 2.3.4.4b virus”

·         Here are a couple references that might bolster the Reference section of this article

·         hpai-livestock-testing-recommendations.pdf (usda.gov)

·         USDA Case Definition Case Definition - Avian Influenza (usda.gov)

Author Response

Reviewer 1

Line 73-74 update to “National Veterinary Services Laboratories (NVSL)”

Answer: Updated as suggested (Line 74-75).

Line 75-76 update to “National Animal Health Laboratory Network (NAHLN)”

Answer: Updated as suggested (Line 76-77).

Edit line 244 to “HP H5N1-IAV 2.3.4.4b virus”

Answer: Edited as suggested (Line 252).

Here are a couple references that might bolster the Reference section of this article

hpai-livestock-testing-recommendations.pdf (usda.gov);

 USDA Case Definition Case Definition - Avian Influenza (usda.gov).: Line 175

Answer: The references have been added as suggested, Line 77 and 175

Reviewer 2 Report

Comments and Suggestions for Authors

The article “Detection and monitoring of highly pathogenic influenza A virus 2.3.4.4b outbreak in dairy cattle in the United States” aims to investigate the presence of anti-nucleoprotein (NP) antibodies in serum and milk and viral RNA in milk on dairy farms affected by outbreaks in USA using a multi-species IAV blocking ELISA and RT-qPCR, also at different stages of the infection.

The paper is quite interesting and well-articulated, showing the efficacy of this ELISA test for the detection of antibody anti-IAV, either in sera or in milk, in particular after the onset of the clinical signs of affected cows. This test could be really useful to monitor the circulation of H5N1-IAV 2.3.4.4b virus in dairy farms.

There are some points to be clarified and better explained:

Line 60: Replace “confirmed” with “suspected”. It’s difficult to diagnose IAV only on the basis of clinical signs, especially in cattle. The symptoms are overlapping with other infectious diseases.

Lines 75-77: What is the NAHLN-approved assay with the deviation to…etc? Please, detail and explain better this part. Write the protocol.

Line 82: The same as lines 75-77

Line 108: Write “room temperature (RT)”.

Lines 110-113: How did you calculate the IAV antibodies values? It’s not described. Did you use a spectrophotometer?  Clarify this part.

Lines 176-177: It’s not clear if 10 or 11 samples were also PCR positive. Please clarify this point.

Table 2: In the caption delete “n=76”. It’s not necessary for the description of this table.

Line 286: Rephrase: “after one week from the onset of clinical signs”.

Author Response

Reviewer 2

The article “Detection and monitoring of highly pathogenic influenza A virus 2.3.4.4b outbreak in dairy cattle in the United States” aims to investigate the presence of anti-nucleoprotein (NP) antibodies in serum and milk and viral RNA in milk on dairy farms affected by outbreaks in USA using a multi-species IAV blocking ELISA and RT-qPCR, also at different stages of the infection.

The paper is quite interesting and well-articulated, showing the efficacy of this ELISA test for the detection of antibody anti-IAV, either in sera or in milk, in particular after the onset of the clinical signs of affected cows. This test could be really useful to monitor the circulation of H5N1-IAV 2.3.4.4b virus in dairy farms.

Answer: We want to thank the reviewer for these supportive words and insightful comments.

There are some points to be clarified and better explained:

Line 60: Replace “confirmed” with “suspected”. It’s difficult to diagnose IAV only on the basis of clinical signs, especially in cattle. The symptoms are overlapping with other infectious diseases.

Answer: This has been corrected as suggested (Line 60).

Lines 75-77: What is the NAHLN-approved assay with the deviation to…etc? Please, detail and explain better this part. Write the protocol.

Answer: This is the RT-qPCR assay described herein (Line 71-87), which was validated, approved by NAHLN network (and USDA) and used by different Veterinary Diagnostic Labororatioes in the USA.

Line 82: The same as lines 75-77

Answer: Done (Line 74-77).

Line 108: Write “room temperature (RT)”.

Answer: “RT” was already defined in line 103.

Lines 110-113: How did you calculate the IAV antibodies values? It’s not described. Did you use a spectrophotometer?  Clarify this part.

Answer: This information has now been added (Line113-114).

Lines 176-177: It’s not clear if 10 or 11 samples were also PCR positive. Please clarify this point.

Answer: We confirm that 11 samples were also positive for PCR (Line 179).

Table 2: In the caption delete “n=76”. It’s not necessary for the description of this table.

Answer: This has been deleted as suggested.

Line 286: Rephrase: “after one week from the onset of clinical signs”.

Answer: This has been rephrased as suggested (Line 293-294).

Reviewer 3 Report

Comments and Suggestions for Authors

The manuscript titled " Detection and monitoring of highly pathogenic influenza A virus 2.3.4.4b outbreak in dairy cattle in the United States" evaluates a commercial ELISA for the detection of H5N1 in US cattle. The manuscript explores pre-clinical, acute and convalescent stages and test both milk and serum sample types. The ELISA results are compared to RT-qPCR. The authors demonstrate the ELISA is an effective tool in detecting antibodies in serum and milk samples from dairy cows and would be an effective tool in the surveillance of H5N1 spread across the US. Overall, the manuscript is well organized and is accessible to a general audience. A minor comment I would like to share is the lack of details on the selection of the particular ELISA kit for this study. The methods section could be strengthened by adding more information on this ELISA such as why the Np antigen was selected for the serological testing and which Ig antibody the kit is detecting. This information would be valuable in supporting the results observed in antibody production over time in the dairy cows testing. 

Author Response

Reviewer 3

The manuscript titled "Detection and monitoring of highly pathogenic influenza A virus 2.3.4.4b outbreak in dairy cattle in the United States" evaluates a commercial ELISA for the detection of H5N1 in US cattle. The manuscript explores pre-clinical, acute and convalescent stages and test both milk and serum sample types. The ELISA results are compared to RT-qPCR. The authors demonstrate the ELISA is an effective tool in detecting antibodies in serum and milk samples from dairy cows and would be an effective tool in the surveillance of H5N1 spread across the US. Overall, the manuscript is well organized and is accessible to a general audience.

Answer: We want to thank the reviewer for these insightful comments.

 A minor comment I would like to share is the lack of details on the selection of the particular ELISA kit for this study. The methods section could be strengthened by adding more information on this ELISA such as why the Np antigen was selected for the serological testing and which Ig antibody the kit is detecting. This information would be valuable in supporting the results observed in antibody production over time in the dairy cows testing. 

Answer: This is a very valid question. Using a multispecies NP ELISA to detect IAV in cattle is justified, as cattle were historically resistant to IAV, with few cases until the emergence of influenza D and highly pathogenic IAV strains. The highly conserved NP protein allows cross-species detection, making the assay practical, cost-effective, and reliable. Evaluation has shown high sensitivity and specificity in cattle, confirming its suitability for early detection and control in this newly recognized host. Specifically for this study, the kit from Idexx because is the one routinely used in our Veterinary Diagnostic Laboratory at Iowa State University.

We have added this information in the Discussion (Line 245-249).

Reviewer 4 Report

Comments and Suggestions for Authors

In their article ” Detection and monitoring of highly pathogenic influenza A virus 2.3.4.4b outbreak in dairy cattle in the United States “ the authors present a methods used for investigation and diagnosis of HPAI in dairy cattle. They estimate a multi-species IAV ELISA and RT-qPCR, and evaluate diagnostic specificity using serum and milk samples.

The provided information is very important and results underscores the efficacy of NP blocking ELISA for detecting and monitoring HPAI H5N1 in dairy cattle.

At the current moment when the influenza virus causes great damage to the poultry industry every year, when it crosses the interspecies barrier and starts infecting mammals, it is very important to have appropriate rapid diagnostic tests. That is why this article is of great importance. I have only a few notes on it:

Materials and methods.

2.1. Samples. When were the samples from farms from Iowa, Montana, South Dakota and Texas taken? During the same period as the others?

2.2. Could you please write the whole name of NVSL?

2.3. Add here the exactly name of the ELISA.

Results.

It is not clear the results of which samples are shown in Figure 2. Add the region and weeks you have tested them.

Line 216. “The diagnostic sensitivity of this antibody test is typically low (<40%) during the first week following the onset of the clinical signs, limiting their primary diagnostic role.” According to whom is low, you have to add cite.

Discussion.

Lines 244-245. “Furthermore, we are developing novel methods to enhance our diagnostic capabilities, including H5-specific antibody detection methods.” To which methods you refer that statement?

Paragraph 2: there is repletion of: “these results”.

The last two paragraphs are more proper for the Conclusion.

Author Response

Reviewer 4

In their article ” Detection and monitoring of highly pathogenic influenza A virus 2.3.4.4b outbreak in dairy cattle in the United States “ the authors present a methods used for investigation and diagnosis of HPAI in dairy cattle. They estimate a multi-species IAV ELISA and RT-qPCR, and evaluate diagnostic specificity using serum and milk samples.

The provided information is very important and results underscores the efficacy of NP blocking ELISA for detecting and monitoring HPAI H5N1 in dairy cattle.

At the current moment when the influenza virus causes great damage to the poultry industry every year, when it crosses the interspecies barrier and starts infecting mammals, it is very important to have appropriate rapid diagnostic tests. That is why this article is of great importance. I have only a few notes on it:

Answer: We much appreciate your comments and suggestions.

Materials and methods.

2.1. Samples. When were the samples from farms from Iowa, Montana, South Dakota and Texas taken? During the same period as the others?

Answer: Yes, these samples were also collected during the Spring of 2024 (Line 65).

2.2. Could you please write the whole name of NVSL?

Answer: Updated as suggested (Line 74-75).

2.3. Add here the exactly name of the ELISA.

Answer: Precise product (catalog) number added in Line 92.

Results.

It is not clear the results of which samples are shown in Figure 2. Add the region and weeks you have tested them.

Answer: Dairy cattle in Kansas. Information updated in Figure 2 caption (Line 211).

Line 216. “The diagnostic sensitivity of this antibody test is typically low (<40%) during the first week following the onset of the clinical signs, limiting their primary diagnostic role.” According to whom is low, you have to add cite.

Answer: A diagnostic sensitivity below 40% can be considered low, which is otherwise normal at the beginning of the infection – a significant number of animals have not yet seroconverted. However, for clarity, we have modified this sentence to reflect the fact that in our study, we found a diagnostic sensitivity below 40% in the first week after the onset of the clinical signs (Line 219).

Discussion

Lines 244-245. “Furthermore, we are developing novel methods to enhance our diagnostic capabilities, including H5-specific antibody detection methods.” To which methods you refer that statement?

Answer: Different immunoassays yet to be defined (Line 253).

Paragraph 2: there is repletion of: “these results”.

Answer: Corrected as suggested (Line 264).

The last two paragraphs are more proper for the Conclusion.

Answer: Agree with the reviewer. We have now combined the last two paragraph of the discussion with the conclusion section (Lines 276-298).